# The Prevalence of Dysphagia in Individuals Living in Residential Aged Care Facilities: A Systematic Review and Meta-Analysis

**DOI:** 10.3390/healthcare12060649

**Published:** 2024-03-13

**Authors:** Hollie Roberts, Kelly Lambert, Karen Walton

**Affiliations:** School of Medical, Indigenous and Health Sciences, University of Wollongong, Wollongong, NSW 2522, Australia; hollie.roberts16@gmail.com (H.R.); klambert@uow.edu.au (K.L.)

**Keywords:** dysphagia, long term care, malnutrition, geriatric, systematic review, meta-analysis

## Abstract

Dysphagia commonly affects older adults, making them nutritionally vulnerable. There is significant variation in the reported prevalence of dysphagia in aged care. The aim of this systematic review and meta-analysis was to determine the prevalence of dysphagia in individuals living in residential aged care facilities using appropriate assessment methods, and in four subgroups at higher risk: individuals with nervous system diseases, dementia, malnutrition, and poor dentition. Scopus, Web of Science, Medline, and CINAHL Plus were searched, and study selection was conducted in Covidence. Meta-analysis using a random effects model was used to obtain the pooled prevalence of dysphagia. Seven studies were eligible for inclusion. Dysphagia prevalence ranged from 16 to 69.6%. The pooled prevalence of dysphagia was 56.11% (95% CI 39.363–72.172, *p* < 0.0001, I^2^ = 98.61%). Sensitivity analysis examining the prevalence of dysphagia using only the CSE indicated a pooled prevalence of 60.90% (95% CI 57.557–64.211, *p* = 0.9994, I^2^ = 0%). Only one study each reported on dysphagia prevalence in individuals with nervous system diseases (31%), poor dentition (92%), and dementia (68.4%), meaning that meta-analysis could not be completed. No studies reported on the prevalence of dysphagia in individuals with malnutrition. The prevalence of dysphagia is high amongst residents of aged care facilities. This evidence should be used to guide improvements in the health outcomes and quality of life of aged care residents. Future research should explore the prevalence in the subgroups at higher risk.

## 1. Introduction

The global population is ageing due to greater life expectancy and sustained lower fertility rates. In 2020, there were 727 million older adults (aged 65 years and over), with projections that this number will double to over 1.5 billion by 2050 [1]. With an ageing population comes an increase in chronic disease, illness, and disability. Consequently, there is high demand for adequate health care systems, including aged care services, that are able to support healthy ageing and ultimately aid in the prevention, management, and treatment of the additional health concerns that are likely to affect these vulnerable individuals [2].

The meagre, and at times, substandard quality of aged care services is a longstanding global issue, with the aged care system often not meeting the minimum standards of care and protection due to demographic, social, and economic challenges [3,4]. This has warranted investigations into the processes of care provided by aged care services such as in the Royal Commission into Aged Care Quality and Safety in Australia [5] and the Care Quality Commission in England [6]. As a result, quality standards for aged care are being modified and progressively regulated, with the allocation of resources and funding finally increasing in some nations. For example, in May 2023, the Australian Government allocated AUD 12.9 million to the establishment of a dedicated Food, Nutrition and Dining Advisory Support Unit in the Aged Care Quality and Safety Commission to ensure that these vulnerable individuals’ nutrition and dining based needs are met, and with dignity and respect [7,8].

Services that provide full-time accommodation, personal care, assistance, and provision of nursing and health care for older individuals, such as residential aged care facilities (RACFs), are required by approximately 4–5% of older adults [9,10]. Of those living in RACFs, older adults represent 95.3% of these residents in Australia [9], 83.1% in the USA [11], and 82.1% in England and Wales [12]. Unfortunately, individuals living in RACFs are nutritionally vulnerable [13,14], with 17.5% of aged care residents reportedly malnourished and 48% at risk of malnutrition [15]. This nutritional vulnerability may be attributable to the reduced functional and cognitive capacity associated with ageing [16], disability, morbidity [17], poor dentition [18,19], age-related decrease in appetite [20], polypharmacy [21], variable food service quality [22], poor mental health, and conditions that affect nutrient and energy intakes such as dysphagia [23].

Dysphagia refers to the difficulty or inability to swallow [24] and can be classified into two major categories: oropharyngeal, which occurs between the oral cavity and upper oesophageal sphincter, and oesophageal, which occurs between the upper and lower oesophageal sphincter [25]. Common signs and symptoms include coughing, choking, poor saliva management, changes in breathing patterns, painful swallowing, and a wet or “gurgly” voice [25]. In older adults, dysphagia often occurs as a result of presbyphagia [26], which refers to the gradual changes in the swallowing mechanism of ageing individuals, in combination with morbidity [27], neurological and neurodegenerative disorders [28,29] such as Parkinson’s disease, dementia, or stroke, structural changes, for example, due to head and neck cancer [30,31], surgery, or trauma, medication use [32] including sedatives, pain relievers, muscle relaxants, or polypharmacy, and muscular conditions such as sarcopenia [33] and achalasia [34]. Irrespective of its cause, dysphagia has detrimental implications [35] including malnutrition [36,37] and unintentional weight loss, nutrient deficiencies, dehydration, frailty, aspiration pneumonia, reduced quality of life (QOL) [38], psychological disorders such as anxiety and depression due to social isolation and anxiety around meal times, increased financial burden on individuals, families, and health care systems, increased rates of hospital admissions [39] and even death.

Dysphagia risk can be identified via screening tools [40,41] such as the EAT-10 questionnaire [42] or the Gugging Swallowing Screen (GUSS) [43]. Dysphagia diagnosis involves the completion of a clinical assessment [44,45], either a non instrumental Clinical Swallow Evaluation (CSE) completed by a speech-language pathologist (SLP) or via validated instrumental assessment. A CSE generally includes obtainment of patient history, patient-reported measures, oral sensorimotor exam, assessment of communication and cognition function, swallow trials, and observations; however, this can vary [46,47]. Instrumental assessment is considered gold standard and includes the Flexible Endoscopic Evaluation of Swallowing (FEES) or Videofluoroscopic Swallow Study (VFSS), otherwise known as the Modified Barium Swallow (MBS) [36]. Despite the availability of valid and reliable methods, dysphagia remains underrecognized and undertreated in aged care settings [41]. This is due to the paucity of clinical governance for routine screening and assessment; infrequent referral to, reimbursement of, and involvement of allied health professionals including dietitians and speech-language pathologists; the inconsistent use of validated screening tools for risk assessment due to the unavailability of an internationally standardized tool [48]; the increased cost and actions associated with instrumental assessment and lack of awareness among residential aged care staff to detect signs of dysphagia and conduct accurate screening [49].

The prevalence of dysphagia, specifically in this population and setting, may be underreported due to heterogenous methods for assessing and diagnosing dysphagia, variations in assessors, the availability of specialist staff, and the limited number of studies. For example, a prospective-observational study conducted in Europe and Israel observed dysphagia in 30.3% [50] of aged care residents via a CSE completed by trained residential aged care staff, whilst a cross-sectional study conducted in Europe and North America reported a prevalence of only 13.4% [51] based on medical records and staff perception. The limited research available also suggests a higher frequency of dysphagia in particular subgroups residing in RACFs, including those with neurological and neurodegenerative disorders (hereon referred to as nervous system diseases) [52], dementia, malnutrition [53], and poor dentition [54,55].

A systematic literature review (SLR), published in 2022, on the prevalence of dysphagia in different healthcare settings estimated that the pooled prevalence of dysphagia in residential aged care was 50.2% [56]. However, this review had several limitations, including the use of only two databases, exclusion of non-English studies, and an estimated prevalence based on only three studies that used screening tools to diagnose dysphagia, thus creating uncertainty in the result. Consequently, further research including additional databases and a modified search strategy restricted to appropriate methods of assessment, specifically clinical assessment, was needed to confirm the prevalence of dysphagia in RACFs.

Quantifying the prevalence of dysphagia in RACFs and in key subgroups is imperative to accurately capture the potential burden of the condition and develop strategies to improve the health outcomes and QOL of these individuals. This evidence can be used to guide prioritization for intervention and the adequate allocation of resources and funding, as well as to increase awareness and education of residential aged care staff and policymakers. Additionally, it can be used to advise advancements in clinical governance, advocate for more allied health professionals, including speech-language pathologists and dietitians to ensure accurate identification, diagnosis, and treatment, alongside advancements in food service management.

Given the limitations and variation in the reported prevalence of dysphagia, the aim of this systematic review (SR) and meta-analysis was to synthesize the available evidence to determine the prevalence of dysphagia in individuals living in RACFs using appropriate assessment methods. A secondary objective was to describe the prevalence of dysphagia in four subgroups at higher risk: individuals living in RACFs with nervous system diseases, dementia, malnutrition, and poor dentition.

## 2. Materials and Methods

The specific question to be answered was “What is the prevalence of dysphagia (concept) in individuals who live in residential aged care facilities (context) (population)?”. The study protocol was registered with the international prospective register of systematic reviews; PROSPERO (CRD42023401857). This SR was reported according to the Preferred Reporting Items for Systematic Reviews and Meta-Analyses (PRISMA) 2020 checklist [57].

### 2.1. Information Sources

A systematic literature search was conducted on 28 March 2023 by one author (H.R.) on four electronic databases: Scopus, Web of Science (Core Collection), Medline, and CINAHL Plus, with no restrictions. The key concepts of dysphagia, assessment, residential aged care facilities, nervous system diseases, dementia, malnutrition, and dentition were searched as keywords and Medical Subject Heading Terms (MeSH), as per the database. The search terms were developed by one author (H.R.) in collaboration with a research librarian, and a preliminary search was conducted on Scopus. The search terms and search strategies for each database are listed in Appendix A, respectively.

### 2.2. Study Selection

Study selection was conducted in Covidence (Systematic Review Software, www.covidence.org, accessed on 11 July 2023) using the eligibility criteria. Duplicates were removed, and title and abstract screening were completed in duplicate by the three authors (H.R. and K.W. or K.L.). Discrepancies were discussed and uncertainties were sent to full text review. One author (H.R.) independently reviewed the full texts, and uncertainties were discussed until consensus was reached between the three authors (H.R., K.L., and K.W.).

Studies were considered eligible for inclusion if they were (i) primary research articles, (ii) reported the prevalence of dysphagia, (iii) diagnosed dysphagia using appropriate assessment methods, that is, via a validated instrumental assessment tool such as the VFSS, MBS, or FEES, or via non instrumental CSE conducted by a qualified health professional such as an SLP or Doctor, or as documented in the medical records if the original diagnosis was made using an appropriate assessment method, and (iv) conducted in RACFs or equivalent. Studies that reported the prevalence of dysphagia in the subgroups of interest (individuals living in RACFs with nervous system diseases, dementia, malnutrition, or poor dentition) were also included. Studies were excluded if they (i) did not report on the outcome of interest (prevalence of dysphagia), (ii) diagnosed dysphagia using inappropriate assessment methods such as screening tools, subjective measures, or a CSE completed by anyone other than a qualified health professional such as an SLP or Doctor, (iii) were conducted in non-residential aged care settings such as hospitals or rehabilitation centres, and (iv) if malnutrition was diagnosed via non validated assessment tools. Studies that utilized populations with previously identified dysphagia symptoms, deemed at risk of dysphagia via screening tools or subjective measures, or focused solely on tube-fed individuals were excluded.

### 2.3. Data Extraction and Summary Measures

One author (H.R.) completed data extraction in Microsoft Word (Version 16.16.27 2016). Each extraction was reviewed by a second author (K.W. or K.L.) for data completeness. The collected data were tabulated and consisted of study demographics (author, year of publication, study location, and study type), population characteristics (sample size, age, and gender ratio), prevalence data for dysphagia and subgroups, the method used in the assessment of dysphagia, and who conducted the assessment. Details on the prevalence of dysphagia in subgroups of interest (individuals with nervous system diseases, dementia, malnutrition, and poor dentition) were extracted if present and summarized in a descriptive format. In addition, information deemed relevant for further context of the results was also extracted and included as comments. Prevalence data were collected as both a fraction and percentage of the population. When data were not reported in this format, the percentage and/or fraction was calculated by one author (H.R.) for homogeneity in reporting of the results. Where outcome data were not available, the author (H.R.) wrote “Not reported” or “Unclear”, and no attempts were made to contact the authors of the study, typically due to the age of the studies.

### 2.4. Quality Appraisal

Quality appraisal of the included studies was completed using the Academy of Nutrition and Dietetics Criteria Checklist for primary research [58]. The checklist uses 10 validity questions based on the research question, subject selection, comparability of study groups, withdrawals, consideration of limitations and bias from blinding, funding and sponsorships, interventions, measurements and outcomes, and statistical analysis to assess study quality. The answers for each question are categorized as Yes, No, Unclear, or Not Applicable, with the final outcome classified as negative, neutral, or positive. Appraisal was completed independently and in duplicate (H.R. and K.W. or K.L.), and uncertainties were discussed until agreement. All studies were included in this review irrespective of quality; however, the limitations of the studies rated as neutral were considered when discussing the results.

### 2.5. Results Synthesis

Data on the number of individuals living in RACFs and the prevalence of dysphagia were exported into Microsoft Excel (Version 16.16.27 2016), followed by MedCalc^®^ Statistical Software (Version 20.116) to conduct the meta-analysis. Statistical significance was set at *p* < 0.05. The degree of variance and heterogeneity between the studies was evaluated using the I^2^ statistic, with a value closer to 100% indicating a higher level of heterogeneity. Due to the variation in methods used for dysphagia assessment, and to reduce variation in the outcome, a random effects model was used to obtain the pooled prevalence of dysphagia. Publication bias was assessed using the Egger’s test.

## 3. Results

The initial search yielded 6329 studies, prior to 2027 duplicates being removed. The titles and abstracts of the remaining 4302 studies were screened, and 154 studies were eligible for full text review. A further 146 studies were excluded, resulting in eight papers describing seven studies eligible for inclusion, as shown in Figure 1. The primary reasons for exclusion were the incorrect outcome being reported and the incorrect assessment method used to obtain diagnosis.

### 3.1. Participants

Four studies were cross-sectional [59,60,61,62,63] and three studies were cohort [64,65,66]. The oldest study was published in 1996 [64] and the most recent study in 2018 [59,62,63]. The studies were geographically distributed across four continents and five countries, with two studies conducted in South America (Brazil [59,62,63]), two studies conducted in North America (USA [64,65]), two studies conducted in Europe (The Netherlands [60] and Spain [66]), and one study conducted in Asia (China [61]). Figure 2 provides an overview of the geographic distribution of the included studies [67].

A total of 3772 participants living in RACFs were included in the review, ranging from 41 to 2384 subjects [65,66] per study. Table 1 outlines the seven studies. The mean age was reported in four studies, ranging from 73.5 ± 8.9 years [62,63] to 88.7 ± 6.8 years [66], whilst three studies did not report the mean age. The mean age of individuals with dysphagia was reported in three studies, ranging from 82.2 ± 9.1 years [61] to 89.4 ± 6.3 years [66]. Four studies did not report the mean age of individuals with dysphagia. The gender ratio was reported in six studies, with the lowest percentage of males at 26.6% [66] and the highest percentage of males at 100% [5]. One study did not report on the gender ratio.

### 3.2. Assessment Tools Used

A variety of assessment methods were used to diagnose dysphagia. A CSE was used to obtain a diagnosis of dysphagia in three studies [59,61,62,63]. Two studies noted results in medical records, with a CSE reported as the original assessment tool [60,66]. One study used a CSE and VFSS to diagnose dysphagia [64], and another study used a CSE and VFSS for all subjects, and a scintigraphy examination and FEES for approximately half of the subjects [65].

### 3.3. Who Completed the Assessment

With regard to who completed the assessment, an SLP conducted the assessment in two studies [59,62,63] and an SLP, alongside trained research assistants including SLP students, conducted the assessment in one study [61]. For the two studies that reported from medical records, one study reported that the original assessment was conducted by an SLP [60] and the other study reported that the original assessment was conducted by a Doctor [66]. One study reported that an SLP conducted one half of the assessment, but did not clarify who conducted the other half; however, it was likely the same SLP [64]. One study did not clarify who conducted the assessment; however, it is likely that it was an SLP [65].

### 3.4. Prevalence of Dysphagia

The prevalence of dysphagia ranged from 16% [60] to 69.6% [66]. In total, seven studies of 3772 residents were included in the meta-analysis [59,60,61,62,63,64,65,66]. The pooled prevalence of dysphagia in individuals living in RACFs was 56.11% (95% CI 39.363–72.172, *p* < 0.0001, I^2^ = 98.61%) (Figure 3). Sensitivity analysis was conducted with the study by Hollaar et al. (2017) [60] due to reporting a significantly lower prevalence of dysphagia. Removing this study from the meta-analysis had an appreciable effect on the pooled prevalence of dysphagia, increasing the prevalence to 64%, and a significant reduction in heterogeneity (95% CI 59.194–68.827, *p* = 0.0007, I^2^ = 76.43%) (Figure 4). Based on the results of the Egger’s test, there was no publication bias for the included studies regarding the prevalence of dysphagia (*p* = 0.4198).

When studies were analysed according to those who solely used a CSE [59,61,62,63] to assess dysphagia, the pooled prevalence of dysphagia was 60.90% (95% CI 57.557–64.211, *p* = 0.9994, I^2^ = 0%). This is shown in Figure 5.

Three studies reported on the prevalence of dysphagia in the subgroups of interest [60,62,63,66], and four studies did not [59,61,64,65]. Meta-analysis could not be completed for the subgroups of interest, as only one study each could be included in three of the subgroups. Regarding the subgroups, one study each reported on the prevalence of dysphagia in individuals with nervous system diseases (31%) [60], individuals with poor dentition (partial and nonfunctional dentition) (92%) [62,63], and in individuals with dementia (68.4%) [66]. No studies reported on the prevalence of dysphagia in individuals with malnutrition; however, one study reported the prevalence of dysphagia in individuals who were underweight (using a BMI ≤ 24.9) (47.3%) [62].

### 3.5. Quality Appraisal

Overall, the quality of the evidence was rated as positive [61,66] or neutral [59,60,62,63,64,65]. Of the eight papers describing the seven studies, six papers had potential bias in subject selection [59,60,62,63,64,65], five were unclear in the comparison of study groups [59,62,63,64,65], two were unclear on the methods of withdrawal [64,65], two did not disclose withdrawal methods [59,62], seven were unclear in the use of blinding [59,60,61,62,63,64,65], six were unclear in the details of the intervention [60,61,62,63,64,65], three were unclear in the use of statistical analysis [63,64,65], two were unclear when considering bias and limitations [65,66], and three had potential bias due to funding or sponsorship [61,64,65]. Appendix A provides further details of the quality assessment.

## 4. Discussion

To our knowledge, this is the first SR and meta-analysis reporting on the prevalence of dysphagia in individuals living in RACFs using appropriate assessment methods, and in subgroups at higher risk, including those with nervous system diseases, dementia, malnutrition, and poor dentition. The prevalence of dysphagia in individuals living in RACFs ranged from 16 to 69.6% [60,66]. Meta-analysis of the seven eligible studies determined that the pooled prevalence of dysphagia was 56.11%. This review, therefore, suggests that the prevalence of dysphagia in this population and setting is high. The prevalence of dysphagia in the subgroups of interest, however, could not be estimated via meta-analysis due to a paucity of studies and lack of available data.

This high prevalence rate indicates that a significant number of aged care residents are at high risk of malnutrition, dehydration, aspiration pneumonia, and a reduced QOL. Prioritization of timely assessment, accurate diagnosis, and appropriate management pathways are, therefore, of utmost importance. Despite a prevalence being estimated, it is clear that the true burden of dysphagia in this population remains underestimated. The limited number of studies eligible for inclusion reveals that appropriate diagnostic methods are not being utilized, resulting in limited accurate prevalence data. Additionally, a large proportion of studies utilized the outcomes of screening tools, subjective measures, or a CSE completed by unqualified individuals to confirm diagnosis.

Screening tools are designed to identify the risk of dysphagia, making them invalid tools for diagnostic assessment. Utilizing the Ohkuma dysphagia screening questionnaire, one study reported the prevalence of dysphagia in 24.96% of residents in an aged care facility [68]. This is less than half of the prevalence estimated in this review. Subjective measures including patient-reported measures or observational signs of dysphagia are also invalid diagnostic methods when used in isolation. Patient-reported measures are unreliable, particularly in this population due to the increased rates of cognitive and sensory impairments that may reduce an individual’s capacity to recognize swallowing difficulty and the common belief that swallowing difficulties are a result of the natural process of ageing [69,70,71]. A Canadian study found that 80% of aged care residents that did not self-report symptoms of impaired swallowing, failed dysphagia risk screening [72]. It is also estimated that 55–59% of individuals with dysphagia experience silent aspiration [73], a risk factor for aspiration pneumonia, which is associated with significant morbidity and mortality in aged care residents [74]. Providing no overt signs or symptoms, the detection of silent aspiration is reliant on objective methods, particularly gold standard instrumental assessment, making subjective measures alone inappropriate and unreliable. Like silent aspiration, other symptoms of dysphagia are not always as apparent or easily identified as that of dysphagia. Assessment upon admission, followed by routine screening and assessment in RACFs, irrespective of patient-reported or observational signs of dysphagia, is therefore crucial [29,75]. This will aid in the prevention of unfavourable outcomes, like aspiration and nursing-home-acquired pneumonia [61], thereby increasing life expectancy and improving QOL.

Assessment completion by unqualified individuals, such as nursing home staff, can result in misdiagnosis and the exacerbation of unfavourable outcomes secondary to delayed diagnosis and intervention. Despite this, a survey conducted on the management pathways of nursing home residents with dysphagia revealed that in approximately 25% of nursing homes, untrained and unskilled staff were involved in the screening or assessment of swallowing difficulties [76]. Hence, it is vital that nursing home staff receive sufficient education and training in order to confidently comply with procedures for dysphagia risk identification through the use of validated screening tools, referral to qualified professionals for assessment and treatment, and the accurate implementation of prescribed interventions [77]. Whilst nursing home staff play a crucial role in aged care, it is important to advocate, whenever possible, for the involvement of allied health practitioners, including speech-language pathologists for accurate assessment, diagnosis, and prescription of treatment, and dietitians, for the prevention of nutritional decline following diagnosis [78,79]. Speech-language pathologists and dietitians play a fundamental role in the mitigation of the burden of dysphagia and its complex sequelae, specifically for vulnerable older adults that rely on their aged care facility for complete care. Moreover, an increase in clinical governance is integral to the standardization and regulation of diagnostic and management procedures completed via validated screening tools, appropriate clinical assessment methods, and qualified professionals in this setting. Clinical governance is defined as “*an integrated set of leadership behaviours, policies, procedures, responsibilities, relationships, planning, monitoring and improvement mechanisms that are implemented to support safe, quality clinical care and good clinical outcomes for each consumer*” [80].

This review also found that there are a paucity of studies using validated gold standard instrumental assessment tools including VFSS or MBS, and FEES for the diagnosis of dysphagia. Only two out of the seven studies included in this review utilized a validated instrumental tool. This finding is consistent with a similar SLR looking at the prevalence and methods of assessment of dysphagia in older adults [81], implying that instrumental assessment is not common practice. This may be due to the requirement of specialized training and equipment that is often only available at hospitals, increased costs, travel, staff availability, and feasibility of completion for residents with additional comorbidities [82]. For example, in aged care facilities in Australia, instrumental assessment requires referral of the resident by a Doctor or an SLP to an outpatient clinic such as a hospital [48]. On the other hand, all studies in this review utilized a CSE, likely due to being non-invasive, cost-effective, and time-efficient, making it the preferred method of choice in this population. Future research should employ instrumental assessment when estimating dysphagia prevalence in this setting and explore strategies that may help to overcome the barriers associated with their use in practice.

Despite previous research identifying those individuals with nervous system diseases, including dementia, malnutrition, and poor dentition, they are at higher risk of dysphagia, specifically those living in RACFs due to an increase in the onset of these conditions as one ages. There was a paucity of studies reporting on this prevalence and using appropriate assessment methods. In aged care settings, the prevalence of dysphagia is proposed to be highest in those with neurological conditions [29,41,83] and severe dementia [52,84]. Thus, it was expected that high prevalence rates would be reported for these subgroups within this review. In our review, Hollaar and colleagues [60] estimated that 31% of nursing home residents with dysphagia also had a nervous system disease, and another study identified a positive correlation between dysphagia and neurological disease (*p* = 0.00) [66]. Comparatively, a study reporting on the prevalence of dysphagia in nursing homes in Europe and North America estimated a co-occurrence of dysphagia and neurological disease in only 16.9% of residents [51]. One study in our review also reported that 68.4% of residents with dementia also had dysphagia [66]. Similarly, an observational study conducted in 2022 reported that 69.1% of nursing home residents with dementia also had dysphagia [50]. In this review, Rech and colleagues [62,63] estimated that 92% of residents with dysphagia had poor dentition (partial and nonfunctional dentition). This finding is consistent with the existing literature on the association between poor dentition and dysphagia in the elderly, specifically the reduction in masticatory function in those who are edentulous, resulting in impaired swallowing [29,85,86,87]. Best-practice guidelines for the diagnosis and treatment of dysphagia suggest that structured screening and use of the CSE in these groups is critical [29,75]. In particular, screening of all people over 65 with onset of swallowing disorders, dysphagia-related risk factors, and/or dysphagia-related symptoms and signs should occur and, if over 80, be screened annually [75].

Alarmingly, no studies reported on the prevalence of dysphagia in residents with malnutrition using validated assessment tools, such as the Mini Nutritional Assessment (MNA) or Subjective Global Assessment (SGA) [88]. However, one study did report that the prevalence of dysphagia in individuals who were underweight (BMI ≤ 24.9) was high, at 47.3% [62]. Comparatively, another SR found that the co-occurrence of dysphagia and malnutrition in long-term care residents ranged from 3 to 28%, with 4% of this population having these diagnoses concurrently [89]. However, it is important to note that studies using non validated malnutrition assessment tools and inappropriate methods of assessment for dysphagia diagnosis were included in the determination of these estimates. As malnutrition is both a risk factor for dysphagia and an adverse outcome of dysphagia, early identification via timely and routine assessment using validated tools and appropriate intervention directed by dietitians is essential to reduce the nutritional vulnerability of individuals living in aged care [53,90]. It is vital that future studies report on the prevalence of dysphagia in these high-risk subgroups in RACFs using appropriate assessment methods, particularly in those who are malnourished, to ensure an accurate representation of the true effect size of dysphagia and improve health care planning.

Interestingly, this SLR found no studies that were conducted in the continents of Australia and Africa. The outcomes of this review, therefore, may not be reflective of the burden of dysphagia in RACFs in these two continents. In contrast, an SLR and meta-analysis on the global prevalence of oropharyngeal dysphagia reported that Africa had the highest prevalence of dysphagia, at 64.2%, and Australia had the lowest, at 7.3% [91]. Future studies should be conducted in these two continents to assess whether this prevalence estimate is also reflective of individuals living in RACFs.

However, given the high dysphagia prevalence estimated in this review, it is important to advocate for increased awareness, and the appropriate allocation of resources and funding to guide improvements in the health outcomes and QOL of these vulnerable individuals. It is recommended that standards for clinical governance [80] of dysphagia are developed to ensure the standardization and regulation of timely and routine assessment, using appropriate assessment methods and administered by qualified health professionals [29,75]; speech-language pathologists and dietitians are accessible to aged care services for the accurate assessment and medical management of dysphagia; nursing home staff are educated and trained accordingly, specifically in the identification and screening of dysphagia risk using validated screening tools; barriers to the accessibility of instrumental assessment are broken down; malnutrition is routinely assessed for using validated tools and advancements in food services are made to reduce the nutritional vulnerability of these individuals after diagnosis. Specifically, in Australia, this evidence can be used to guide the actions of the Food, Nutrition and Dining advisory support unit developed under the Royal Commission into Aged Care Quality and Safety and the allocation of the AUD 12.9 million budget.

This review is not without its limitations. Meta-analysis could not be completed in the subgroups of interest, therefore limiting the ability of this review to provide objective conclusions. There was also a high level of heterogeneity between studies when pooled. Inclusion of the study conducted by Hollaar and colleagues [60] appreciably increased the level of heterogeneity, and may be a result of varying assessment methods to determine dysphagia within the study and three different examination and timepoints for assessment between subjects and three nursing homes, as well as an underestimation of prevalence, as identified in the paper itself, resulting in a significantly lower prevalence of dysphagia when compared to the other studies included in this review. The heterogeneity may also be attributed to the variation in assessment methods, assessors, and timeframes for assessment between studies, as well as the substantial difference between the study with the smallest sample size (n = 41) [65] and the largest sample size (n = 2384) [66]. Despite a random effects model being used, this high level of heterogeneity suggests that there is some uncertainty in the results when results are pooled. Additionally, six out of the eight included papers were quality-rated as neutral, which also suggests uncertainty in the results due to potential bias and ambiguity in the methods. Any studies published in languages other than English were not included.

This review also has many strengths, including a strict eligibility criterion that restricted the inclusion of studies to those using appropriate clinical assessment methods. The psychometric validity of the included studies therefore allowed us to provide the most accurate estimation of dysphagia prevalence in RACFs using the available data. This review also followed a rigorous methodology, reporting according to the PRISMA 2020 checklist, and study selection involving deliberation between all three authors (H.R., K.L., and K.W.) until consensus for conflicts and uncertainties.

## 5. Conclusions

Overall, this review suggests that the prevalence of dysphagia in individuals living in RACFs is high (pooled prevalence of 56.11%, but higher at 60.90% when only CSE methods are used). It is strongly recommended that pathways for the timely and routine assessment of dysphagia, using appropriate assessment methods such as the CSE and by qualified health professionals, is prioritized and standardized to prevent and reduce the likelihood of adverse health outcomes and poor QOL in aged care residents. Further research is needed to determine the prevalence of dysphagia in aged care residents with nervous system diseases, dementia, malnutrition, and poor dentition using appropriate assessment methods.

## Figures and Tables

**Figure 1 healthcare-12-00649-f001:**
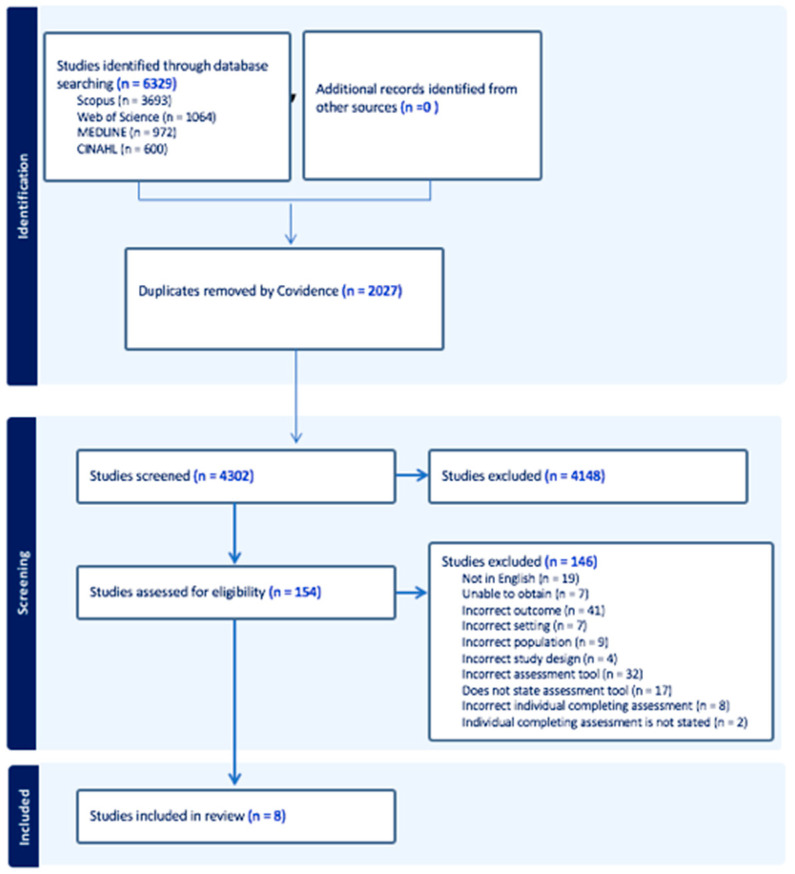
PRISMA 2020 flow diagram on included studies.

**Figure 2 healthcare-12-00649-f002:**
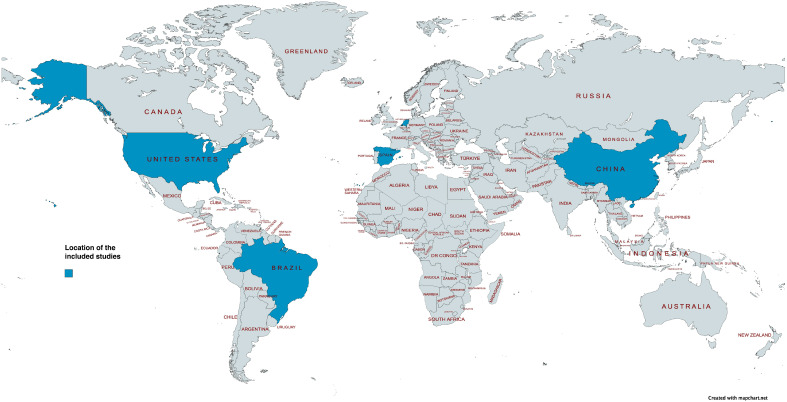
Location of the included studies coloured in blue. Map created with Mapchart.com.

**Figure 3 healthcare-12-00649-f003:**
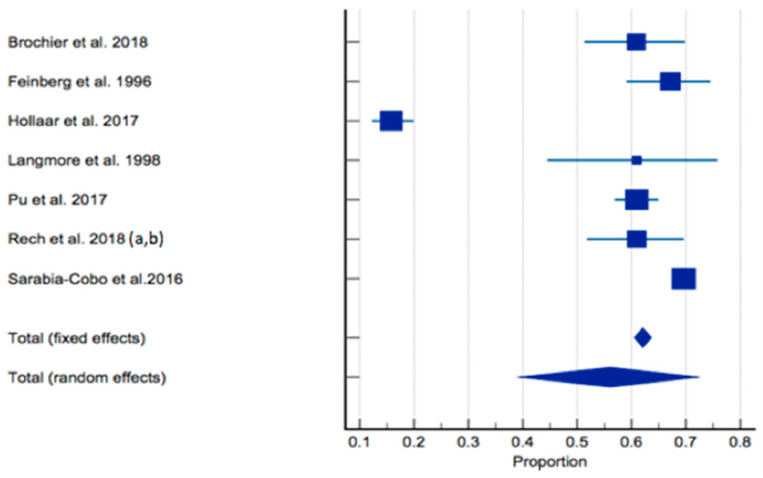
Forest plot of prevalence of dysphagia using random effects model [59,60,61,62,63,64,65,66]. Q = 431.6698, DF 6, *p* < 0.0001, I^2^ = 98.61%, 95% CI for I^2^ = 98.08–98.99. Publication bias—Egger’s test, intercept = −4.9990, 95% CI = −19.6226–9.6247, *p* = 0.4198.

**Figure 4 healthcare-12-00649-f004:**
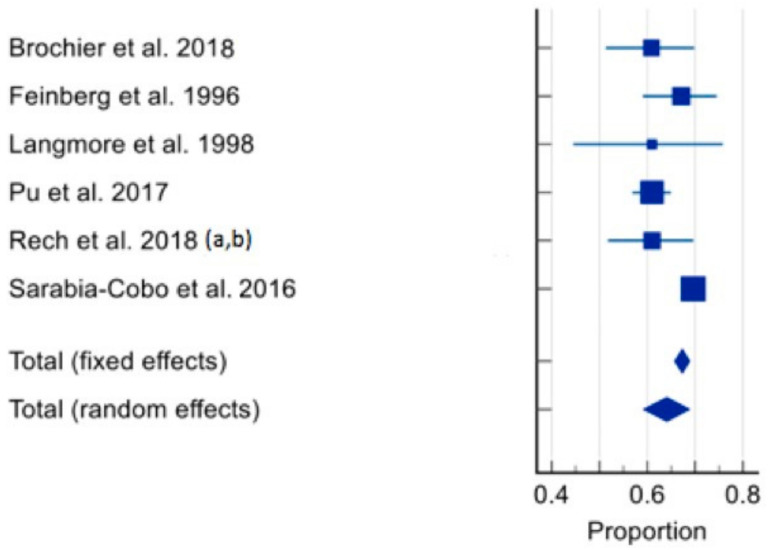
Forest plot of prevalence of dysphagia using random effects model [59,61,62,63,64,65,66] (excluding Hollaar et al. 2017 [60]). Q = 21.2149, DF 5, *p* = 0.0007, I^2^ = 76.43%, 95% CI for I^2^ = 47.22–89.48. Publication bias—Egger’s test, intercept = −2.1602, 95% CI = −5.2598 to 0.9394, *p* = 0.1251.

**Figure 5 healthcare-12-00649-f005:**
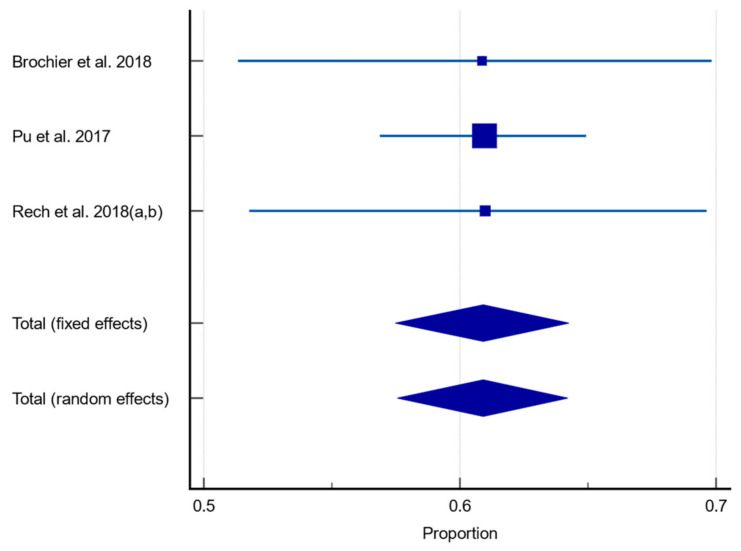
Forest plot of prevalence of dysphagia assessed using CSE only using random effects model [59,61,62,63]. Q = 0.001121, DF 2, *p* = 0.9994, I^2^ = 0%, 95% CI for I^2^ = 0–0. Publication bias—Egger’s test, intercept = −0.04661, 95% CI = −0.3681–0.2789, *p* = 0.3301.

**Table 1 healthcare-12-00649-t001:** Description of the seven included studies.

Author, Year, Country	Study Type	Sample Size (n)	Age (Years)	Gender (% Male)	Prevalence of Dysphagia (n, %)	Tool for Dysphagia Assessment	Professional Who Conducted Dysphagia Assessment	Comments
Brochier et al., 2018, Brazil [59]	Prospective cross-sectional	115 (3 long term care residences)	≥60 Age range: 60–70 (19.1%) 71–80 (36.6%) ≥81 (44.3%)	33% M	70/115 (60.9%) In nervous system diseases: Not reported. In Dementia: Not reported. In Malnutrition: Not reported. In poor dentition: Not reported.	CSE	SLP	Higher xerostomia associated with oropharyngeal dysphagia (*p* < 0.001). Population able to provide informed consent.
Feinberg et al., 1996, USA [64]	Prospective cohort	152 (Single long-term care facility)	Mean age: 86 Age range: 78–96	36.8% M	102/152 (67%) Oropharyngeal dysphagia: 81/102 (79%) Oesophageal dysphagia: 19/102 (18.6%) In nervous system diseases: Not reported. In dementia: Not reported. In malnutrition: Not reported. In poor dentition: Not reported.	VFSSCSE	CSE—SLP VFSS—Unclear, likely SLP	VFSS completed by 52/152 had 0% history of noted dysphagia (i.e., silent aspiration apparent). Method for gaining consent not reported.
Hollaar et al., 2017 The Netherlands [60]	Retrospective cross-sectional	373 (3 nursing homes)	≥65 Mean age with dysphagia: 82.2 ± 9.1 Mean age without dysphagia: 83.5 ± 7.8	30.3% M	59/373 (16%) In nervous system diseases: 18/59 (31%) In dementia: Not reported. In malnutrition: Not reported. In poor dentition: Not reported.	Data obtained from medical records. Original diagnostic tool: CSE upon admission, CSE after a cerebrovascular accident or in case of a history of cerebrovascular accident at the time of admission, CSE after reported clinical symptoms of or complaints about swallowing problems or previously registered signs and symptoms in the electronic medical file before admission.	SLP	Number of diagnostic methods completed by SLP unclear. Where dysphagia diagnosis was unclear, the SLP and elderly care physician were consulted for additional information. Method for gaining consent not reported.
Langmore et al., 1998, USA [65]	Prospective cohort	41 (Single nursing home care centre)	≥60	100% M	25/41 (61.6%) In nervous system diseases: Not reported. In dementia: Not reported. In malnutrition: Not reported. In poor dentition: Not reported.	CSE, VFSS, Scintigraphy examinations (~50% of subjects); Modified FEES (~50% of subjects).	Unclear—likely SLP	81% of subjects with aspiration pneumonia had oropharyngeal dysphagia. Population able to provide informed consent.
Pu et al., 2017, China [61]	Prospective cross-sectional	584 (22 long term care facilities)	≥ 60 Mean age: 84.5 ± 7.9 Mean age with dysphagia: 85 Mean age without dysphagia: 83.6 Age range: 56–105 Median age: 85	Not reported.	356/584 (61.1%) In nervous system diseases: Not reported. In dementia: Not reported. In malnutrition: Not reported. In poor dentition: Not reported.	CSE	SLP and trained research assistants including SLP students	Risk of dysphagia for total sample higher in those with pneumonia (OR 3.578 95% CI: 1.568–8.168, *p* = 0.002). Population able to provide informed consent.
Rech et al., 2018 [62] (a) and Rech et al., 2018 [63] (b), Brazil	Prospective cross-sectional	123 (exact number of long-term care facilities not reported, located in 3 separate towns)	≥60 years Mean age: 73.5 ± 8.9	34.1% M	CSE: 75/123 (61%) EAT-10 tool: 25/123 (20.3%) DenSAT tool: 65/123 (52.8%) In nervous system diseases: Not reported. In dementia: Not reported. In malnutrition: Not reported. In those with partially functional dentition: 14/75 (18.7%). In those with Nonfunctional dentition: 55/75 (73.3%).	CSE, EAT-10 screening questionnaire; DenSAT assessment tool: Developed by 3 × SLP and conducted by dentists. Questionnaire—Subjective dysphagia. Oral motor and functionality assessment; Swallow assessment.	CSE—SLP DenSAT—Dentist	Dysphagia in those with ≥4 oral sensory motor alterations: 54/75 (72%). Only 27.6% reported difficulty swallowing (i.e., silent aspiration apparent). 47.3% of individuals with dysphagia also had a BMI <24.9 (underweight). Population able to provide informed consent.
Sarabia-Cobo et al., 2016, Spain [66]	Prospective cohort	2384 (12 nursing homes)	Mean age: 88.7 ± 6.8 Mean age with dysphagia: 89.4 ± 6.3 Age range: 69–101	26.6%	1659/2384 (69.6%) In nervous system diseases: Not reported. In dementia: 1135/1659 (68.4%) In malnutrition: Not reported. In poor dentition: Not reported.	Data obtained from medical records. Original diagnostic tool: CSE.	Doctor	Positive correlation between the presence of neurological disease and dysphagia (rho = 0.56, *p* = 0.00), cognitive impairment and dysphagia (rho = 0.61, *p* = 0.01) and dysphagia and functional capacity (r = 0.65, *p* = 0.00). Method for gaining consent not reported.

Legend: CSE—Clinical Swallow Evaluation, EAT-10—Eating Assessment Tool, FEES—Fibreoptic Endoscopic Evaluation of Swallowing, SLP—speech language pathologist, VFSS—Videofluoroscopic Swallow Study.

## Data Availability

Data are available on reasonable request.

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
