# Peer review of "The Prevalence of Dysphagia in Individuals Living in Residential Aged Care Facilities: A Systematic Review and Meta-Analysis"

_healthcare, 2024, doi:10.3390/healthcare12060649_

Round 1
Reviewer 1 Report
Comments and Suggestions for Authors
The prevalence of oropharyngeal dysphagia in the elderly community, mainly in Residential Aged Care Facilities, is an important aspect in the public health context of this population considering the possible complications for pulmonary and nutritional conditions resulting from dysphagia. Therefore, I congratulate the authors for choosing the theme and study design. Another relevant aspect in the study refers to the importance of only including studies where oropharyngeal dysphagia was assessed using an appropriate method, assertively excluding studies that only applied dysphagia screening tools which are not indicated for diagnosis.
On the other hand, most of the included studies applied only clinical assessment, which is a method with low accuracy for diagnosing dysphagia with silent aspiration or cases with initial impairment only in the oral phase of swallowing, without signs suggestive of laryngeal penetration or aspiration, if performed by professional without expertise. It is essential that authors understand that choosing studies that mostly used only this method, without FEES or MBS, can compromise prevalence and this should be widely discussed.
However, this reviewer understands that some adjustments and clarifications are necessary to finalize this decision:
1-Aim: I suggest that the authors review the objective since the issue involving the sub-groups concerns the findings of the population of each article included and not pre-selected criteria for the inclusion of the search. Therefore, I suggest that the objective be summarized as "The aim of this systematic review and meta-analysis was to determine the prevalence of dysphagia in individuals living in in residential aged care facilities using appropriate assessment methods."
2-Results: Table 1 must contain the description of the individualse (diagnosis of the population of each article included or even if the authors did not describe the population). It is observed that 3 of the articles were mainly aimed at a dental issue, produced by a group of dentistry specialists and, therefore, the division into a sub-group with poor dentition is a specificity in this context and not a risk group for dysphagia . It is essential that in the 7 articles included, the population is described in table 1. This aspect aims to allow the authors to better discuss not only the prevalence of dysphagia in Residential Aged Care Facilitie, but rather, that the results can clearly show that it is this elderly . Generalizing the prevalence of dysphagia in the context that includes healthy elderly people and those with other etiological diagnoses does not allow assertive actions in these places.
3-Conclusion: To maintain this conclusion " Further research is needed to determine the prevalence of dysphagia in aged care residents with nervous system diseases, dementia, malnutrition, and poor dentition using appropriate assessment methods" it will be necessary for table 1 to contain data from the population included in the studies.
Comments on the Quality of English LanguageThe English Language is very good.
Author Response
Reviewer 1
The prevalence of oropharyngeal dysphagia in the elderly community, mainly in Residential Aged Care Facilities, is an important aspect in the public health context of this population considering the possible complications for pulmonary and nutritional conditions resulting from dysphagia. Therefore, I congratulate the authors for choosing the theme and study design. Another relevant aspect in the study refers to the importance of only including studies where oropharyngeal dysphagia was assessed using an appropriate method, assertively excluding studies that only applied dysphagia screening tools which are not indicated for diagnosis.
On the other hand, most of the included studies applied only clinical assessment, which is a method with low accuracy for diagnosing dysphagia with silent aspiration or cases with initial impairment only in the oral phase of swallowing, without signs suggestive of laryngeal penetration or aspiration, if performed by professional without expertise. It is essential that authors understand that choosing studies that mostly used only this method, without FEES or MBS, can compromise prevalence and this should be widely discussed.
However, this reviewer understands that some adjustments and clarifications are necessary to finalize this decision:
1-Aim: I suggest that the authors review the objective since the issue involving the sub-groups concerns the findings of the population of each article included and not pre-selected criteria for the inclusion of the search. Therefore, I suggest that the objective be summarized as "The aim of this systematic review and meta-analysis was to determine the prevalence of dysphagia in individuals living in residential aged care facilities using appropriate assessment methods."
- Thank you for the suggestion regarding the revised study objective. I have reviewed and discussed this with co-author, Associate Professor Lambert, and we believe that the original objective should remain. The original pre-search PROSPERO registration did list and purposely search terms related to each of the sub-groups.
2-Results: Table 1 must contain the description of the individuals (diagnosis of the population of each article included or even if the authors did not describe the population). It is observed that 3 of the articles were mainly aimed at a dental issue, produced by a group of dentistry specialists and, therefore, the division into a sub-group with poor dentition is a specificity in this context and not a risk group for dysphagia. It is essential that in the 7 articles included, the population is described in table 1. This aspect aims to allow the authors to better discuss not only the prevalence of dysphagia in Residential Aged Care Facilities, but rather, that the results can clearly show that it is this elderly. Generalizing the prevalence of dysphagia in the context that includes healthy elderly people and those with other etiological diagnoses does not allow assertive actions in these places.
- Thank you for these comments for consideration. Table 1 provides details describing the individuals in the included studies. All the articles included people (aged 60 yrs or over), who were living in residential aged care with dysphagia. Although 3 articles also referred to dental issues (a sub-group of interest in our review), each of the articles still contained data on the prevalence of dysphagia for the group studied, which was still the primary outcome of interest.
3-Conclusion: To maintain this conclusion " Further research is needed to determine the prevalence of dysphagia in aged care residents with nervous system diseases, dementia, malnutrition, and poor dentition using appropriate assessment methods" it will be necessary for table 1 to contain data from the population included in the studies.
- Table 1 contains a summary of the available data. However, the small number of available studies exploring these sub-groups, coupled with the high-risk nature of dysphagia on nutritional health and the variable study quality means that the authors still feel that this conclusion is relevant.
Reviewer 2 Report
Comments and Suggestions for Authors
Thank you for this interesting and important paper. I have the following questions and comments:
1. Could you expand further on the Hollaar et al study as an outlier?
2. It may be informative to know the breakdown of the incorrect outcome (n = 41) and tool results (n=32), which may provide supplementary information towards the topic.
3. How reasonable is it to expect 100% of RACF facilities to conduct VFSS, MBS, or FEES on all patients?
4. Could you compare the meta analysis results of those "gold standard" measures to other assessment methods and risk methods? Such as EAT-10, GUSS, or other assessments/instruments? In this way, we could get a gold standard comparison against other more practical measures or "cruder" instruments to know what the "gap" is in how we are underestimating dysphagia. This provides support for increasing use of the gold standard, but also knowing what the lesser methods are telling us.
5. Are there more characteristics of the RACF's that could be expanded upon?
None
Author Response
Thank you, I have uploaded a file due to the additional Figure (5) mentioned.

Reviewer 3 Report
Comments and Suggestions for Authors
Although it is generally known that elderly people living in nursing homes who require care are at high risk of developing dysphagia, it is surprising that the incidence rate has not been accurately determined.
Examining the prevalence of dysphagia in nursing home residents can help us understand their potential risks. Therefore, I think this research is valuable. However, there are some biases and expressions that are difficult to understand, so I recommend that they be revised.
Line 314-315 “Despite a prevalence … population remains underestimated.”
Why did the authors think this way? It may be short-sighted to think that dysphagia is being underestimated just because the treatment for dysphagia is inadequate. Consideration should be given to the circumstances in which dysphagia cannot be dealt with even if swallowing disorders are considered important, such as economic and social reasons, a lack of specialists, and lack of medical skills.
Line 342-347 “Assessment completion by … of swallowing difficulties [76].”
This can be interpreted as saying that it is not desirable for medical staff who are not experts to evaluate swallowing function. It is impossible for experts to intervene in all nursing homes. I believe that in facilities where there are no specialists available to request swallowing evaluations, it is worthwhile to have non-specialist staff perform swallowing function evaluations. I agree with the authors' opinion that there should be more opportunities for experts to evaluate swallowing, but consideration needs to be given to facilities that cannot rely on experts.
Line 357 What exactly does “clinical governance” refer to? Please add a description.
Author Response
Reviewer 3
Although it is generally known that elderly people living in nursing homes who require care are at high risk of developing dysphagia, it is surprising that the incidence rate has not been accurately determined.
Examining the prevalence of dysphagia in nursing home residents can help us understand their potential risks. Therefore, I think this research is valuable. However, there are some biases and expressions that are difficult to understand, so I recommend that they be revised.
The In-text changes are indicated below in italics.
Line 314-315 “Despite a prevalence … population remains underestimated.”
Why did the authors think this way? It may be short-sighted to think that dysphagia is being underestimated just because the treatment for dysphagia is inadequate. Consideration should be given to the circumstances in which dysphagia cannot be dealt with even if swallowing disorders are considered important, such as economic and social reasons, a lack of specialists, and lack of medical skills.
- The comments have been considered and the paper adjusted to say that dysphagia “may be under-reported” rather than “is likely under-reported”
- The introductory section is shown in the manuscript with track changes, but now reads as follows:
“The prevalence of dysphagia, specifically in this population and setting, may be under-reported due to heterogenous methods for assessing and diagnosing dysphagia, variations in assessors, the availability of specialist staff and the limited number of studies.”
- A further recent meta-analysis in response to reviewer 2’s comments (as explained above regarding Figure 5) suggests further evidence of under reporting. In inclusion of only studies using the CSE [60, 62-64] found a higher pooled prevalence of dysphagia was 60.90 % (95% CI 57.557 -64.211, p =0.9994, I2= 0%).
Line 342-347 “Assessment completion by … of swallowing difficulties [76].”
This can be interpreted as saying that it is not desirable for medical staff who are not experts to evaluate swallowing function. It is impossible for experts to intervene in all nursing homes. I believe that in facilities where there are no specialists available to request swallowing evaluations, it is worthwhile to have non-specialist staff perform swallowing function evaluations. I agree with the authors' opinion that there should be more opportunities for experts to evaluate swallowing, but consideration needs to be given to facilities that cannot rely on experts.
- These pragmatic comments have also been considered. The authors were aiming for the most rigorous approach for this SLR and meta-analysis. However, it is noted that these specialist staff and particular assessments may not always be available. This was evidenced by the already included 2022 reference [78] that surveyed residential aged care facilities nationally in Norway to investigate the management of dysphagia, finding that residents in approximately 75% of residential aged care facilities were not routinely screened or assessed for dysphagia.
- I do think that the following existing text speaks to this issue:
“Hence, it is vital that nursing home staff receive sufficient education and training in order to confidently comply with procedures for dysphagia risk identification through the use of validated screening tools, referral to qualified professionals for assessment and treatment, and the accurate implementation of prescribed interventions [79].”
- However, the manuscript has been updated to further acknowledge the point made, with the following wording:
“Whilst nursing home staff play a crucial role in aged care, it is important to advocate, whenever possible, for the involvement of allied health practitioners, including speech-language pathologists for accurate assessment, diagnosis and prescription of treatment, and dietitians, for the prevention of nutritional decline following diagnosis [80,81].”
Line 357 What exactly does “clinical governance” refer to? Please add a description.
- Thank you for this feedback. Clarity has been improved by including the following description in brackets after “Clinical Governance….”.
- The Australian Government Aged Care and Quality Safety Commission definition of clinical governance is now included with the appropriate reference.
- Clinical governance is defined as, “an integrated set of leadership behaviours, policies, procedures, responsibilities, relationships, planning, monitoring and improvement mechanisms that are implemented to support safe, quality clinical care and good clinical outcomes for each consumer [82].”
A further note
On review the authors noted that reference to nursing homes and residential aged care had been used interchangeably in the text. For consistency, all references have been changed to residential aged care.
Round 2
Reviewer 1 Report
Comments and Suggestions for Authors
Dear authors,
This reviewer agrees with the researchers about the importance of understanding who the population was in this type of study. There are two ways to achieve this: describe all the profiles found in each study or pre-define (as was done in the study by the authors) the groups in which these etiological variables will be described. Therefore, it is essential that the authors mention in the data extraction method how this population will be grouped (in table 1 the interest in identifying whether the poor dentition of this population is mentioned is clear, and mixes with the description of the underlying neurological etiologies for dysphagia ). It is imperative that authors define the search for the groups analyzed in the included studies on the extratinon date. How to point out in the results what had not been pre-defined? After this inclusion in the method, I believe it will make more sense to mention that no study has described poor dentition. It is important to emphasize to the authors that poor dentition in geriatrics is not an isolated factor for dysphagia but rather for impaired chewing.
Author Response
Thank you for the further feedback. The authors are not entirely clear on the question being asked but additional text has been added in the methods section as follows (lines 181-184): Details on the prevalence of dysphagia in subgroups of interest (individuals with nervous system diseases, dementia, malnutrition, and poor dentition) were extracted if present and summarized in a descriptive format.
Reviewer 2 Report
Comments and Suggestions for Authors
If further reviewer comments are warranted, I would request that the authors include a discussion on the prevalence of dysphagia in the original analyses vs. the new Figure 5 analyses. Are they similar or different? What are the implications, or lack thereof, for clinical practice, screening, etc?
Comments on the Quality of English LanguageNone
Author Response
Thank you for the further review and comments. The authors have included the following wording in the following sections that address the point raised.
Abstract (Lines 19-20): Sensitivity analysis examining the prevalence of dysphagia using only the CSE indicated a pooled prevalence of 60.90 % (95% CI 57.557 -64.211, p =0.9994, I2 = 0%).
Results (Lines 276-278): When studies were analyzed according to those who solely used a CSE [60, 62-64] to assess dysphagia, the pooled prevalence of dysphagia was 60.90 % (95% CI 57.557 -64.211, p =0.9994, I2 = 0%). This is shown in Figure 5.
Conclusion (Lines 492-493): Overall, this review suggests that the prevalence of dysphagia in individuals living in RACF is high (pooled prevalence of 56.11%, but higher at 60.90% when only CSE methods are used).